# The Video Microscopy-Linked Electrochemical Cell: An Innovative Method to Improve Electrochemical Investigations of Biodegradable Metals

**DOI:** 10.3390/ma14071601

**Published:** 2021-03-25

**Authors:** Tycho Zimmermann, Norbert Hort, Yuqiuhan Zhang, Wolf-Dieter Müller, Andreas Schwitalla

**Affiliations:** 1Dental Materials and Biomaterial Research, Department of Prosthodontics, School of Dentistry, Charité-University Medicine Berlin, Aßmannshauser Str. 4–6, 14197 Berlin, Germany; tycho.zimmermann@charite.de (T.Z.); yuqiuhan.zhang@charite.de (Y.Z.); andreas.schwitalla@charite.de (A.S.); 2Helmholtz Zentrum Geesthacht, 21502 Geesthacht, Germany; norbert.hort@hzg.de

**Keywords:** magnesium, biomaterials, electrochemistry, corrosion, degradation, hydrogen evolution, microscopy

## Abstract

An innovative, miniature video-optical-electrochemical cell was developed and tested that allows for the conducting of electrochemical corrosion measurements and simultaneous microscopic observations over a small, well-defined surface area of corroding or degrading samples. The setup consisted of a miniature electrochemical cell that was clamped onto the metal sample and fixed under a video microscope before being filled with electrolyte. The miniature cell was comprised of afferent/efferent electrolyte ducts as well as a connection to the Mini Cell System (MCS) for electrochemical measurements. Consequently, all measured and induced currents and voltages referred to the same small area corroding completely within the field of view of the microscope, thus allowing for real-time observation and linking of surface phenomena such as hydrogen evolution and oxide deposition to electrochemical data. The experimental setup was tested on commercial purity (cp) and extra-high purity (XHP) magnesium (Mg) samples using open circuit potential and cyclic voltammetry methods under static and flowing conditions. The corrosion potential was shifted more anodically for cp Mg in comparison to XHP Mg under dynamic conditions. The corrosion current assessed from the cyclic voltametric curves were higher for the cp Mg in comparison to XHP Mg. However, there were no differences between static and flow conditions in the case of XHP Mg in contrast to cp Mg, where the current density was two times higher at dynamic conditions. The measurements and observations with this new method pave the way for a more detailed understanding of magnesium corrosion mechanisms, thus improving predictive power of electrochemical corrosion measurements on newly developed magnesium or other biodegradable alloys applied for medical devices. Different electrochemical tests can be run under various conditions, while being easy to set up and reproduce as well as being minimally destructive to the sample.

## 1. Introduction

Magnesium displays promising potential as implantable biodegradable biomaterial for cardiovascular stents, bone fixation devices, and tissue engineering scaffolds and has been an object of intensive research [1]. Apart from being biodegradable, it is an important trace mineral in human organisms and, thus, fully biocompatible in its unalloyed form [2,3,4]. Furthermore, magnesium displays a favorable bone response [5,6]. Among implant metals, the elastic modulus of magnesium is the most similar to that of bone (41–45 GPa for magnesium vs. 3–20 GPa for bone [7]), thus minimizing stress shielding [3,8].

The predominant drawbacks of magnesium when used as a biomaterial relate to its corrosion characteristics. In its unalloyed form, magnesium displays relatively rapid, localized, and difficult-to-predict degradation when implanted. The unique environment of the human body, comprising physiological temperature, perfusion, abundance of inorganic ions, amino acids, proteins, and buffering agents, affects the degradation rate and mechanism in as yet unpredictable ways [9]. This may lead to premature failure of the implanted magnesium structures bringing about grave ramifications to patients’ health [10].

Furthermore, magnesium produces large volumes of hydrogen gas during corrosion. If rapid corrosion occurs, the hydrogen promotes the accumulation of gas bubbles in the tissue surrounding the implant site [11]. Because of the degradation reaction, local pH values are increased in the tissue surrounding the implant. This can negatively affect physiological reactions [7].

Current scientific efforts aim for the goal of slowing down magnesium corrosion and achieving a more uniform and predictable corrosion mode, while at the same time improving mechanical properties and host response [12]. Common methods include alloying [13,14], grain refining, surface modifications, and coating with layers of absorbable materials such as calcium phosphate or biocompatible polymers [15,16,17,18,19,20]. However, each of these methods comes with its own set of challenges, first among which is a decline in biocompatibility [21]. Whereas countless promising advancements have been made in the lab, only few magnesium-based implants have made it to the stage of animal and human trials. A further problem standing in the way of clinical translation is the difficulty of accurately predicting in vivo magnesium degradation through in vitro testing [22,23], in part due to the absence of a commonly accepted methodology [23]. This leads to a wide range of magnesium alloys having to be considered for costly and ethically challenging animal trials [9]. So far, only a few magnesium-based products are commercially available in the form of magnesium stents made of proprietary magnesium alloy (Biotronik SE & Co. Kg, Berlin, Germany) and orthopedic screws made of MgYREZr (Syntellix AG, Hanover, Germany) [1]. 

The dissolution of magnesium in aqueous media is described by the following chemical reactions [24,25,26]:

Anodic reaction:(1)$$\mathrm{Mg}\to {\mathrm{Mg}}^{2+}+\;2e^{-}$$

Cathodic reaction:(2)$$2H_{2}O\;+\;2e^{-}\to 2{\mathrm{OH}}^{-}+{\;H}_{2}↑$$

Overall reaction:(3)$$\mathrm{Mg}+2H_{2}O\to {\mathrm{Mg}}^{2+}+2{\mathrm{OH}}^{-}+H_{2}$$

Product formation:(4)$${\mathrm{Mg}}^{2+}+\;2{\mathrm{OH}}^{-}\to \mathrm{Mg}{\left(\mathrm{OH}\right)}_{2}$$

Hydrogen gas can be observed developing on the surface of corroding magnesium samples in the form of gas bubbles.

Mg(OH)_2_ is not soluble and forms a partially protective layer on the magnesium or magnesium alloy surface [25]. The stability of this layer is dependent on local pH value. In neutral and alkaline media, this layer is porous and only offers limited protection [24,26]. In the presence of chlorides in a concentration of more than 30 mmol/L, Mg(OH)_2_ converts to soluble MgCl_2_ [25,27,28]. This is the case in body fluids, where a chloride concentration of about 150 mmol/L exists. Other inorganic ions found in body fluids such as phosphate and carbonate further complicate the corrosion process [7,9].

The corrosion mechanisms and corrosion speed are strongly influenced by the components, surface and bulk treatment of the respective alloy, and by trace impurities in pure magnesium samples [12,24,26,28,29].

In physiological media, magnesium usually corrodes in the form of pitting corrosion or extremely localized corrosion, as opposed to general corrosion [9,26,30]. These corrosion modes are the most problematic for structural integrity and predictability. 

In vitro corrosion measurement methods play an important role in choosing among the wide variety of magnesium alloys and surface modifications those with the most potential for further development and in vivo testing. Furthermore, in vitro methods allow for the elucidation of the particular corrosion mechanisms displayed by different modifications.

The most common in vitro corrosion testing methods for magnesium include hydrogen collection measurements, weight/volume loss measurements, and electrochemical testing. Each of these test methods comes with its unique set of challenges [31,32,33].

In practice, the hydrogen collection is far from efficient and the ratio of mass loss as estimated by hydrogen collection to actual mass loss is much lower than can theoretically be expected. In a review [31], the range of reported ratios was found to lie between 0.22:1 and 1.31:1. This was attributed to a number of reasons, mainly concerning experimental setup.

In mass loss experiments, the correct removal of corrosion product is crucial to obtain a valid result. In the absence of an established standard of conducting mass loss measurement on magnesium, and in recognition of the fact that results are highly dependent on experimental setup, especially sample surface-to-electrolyte volume ratio, a wide range of corrosion rates has been reported [31].

Electrochemical methods (ECM) permit meaningful data to be gleaned by perturbing the relative equilibrium at the surface of the degrading sample by electrically charging the sample surface in the cathodic or anodic direction and measuring the resulting current. Commonly employed measurement protocols include potentiodynamic polarization (PDP), in which a range of DC (dircet current) potentials’ cathodic and anodic to the OCP (Open circuit potential) is applied. This permits the isolation and interpretation of the kinetics of cathodic and anodic processes. The voltammetric potentials can be applied as a linear sweep (LSV) or in repeating cycles (CV) and result in PDP plots of I (current) vs. E (potential). These plots provide information on cathodic, anodic, and corrosion current densities as well as corrosion potentials [31,34,35,36].

Another commonly employed method is electrochemical impedance spectroscopy (EIS) [37,38,39,40]. The choice of the correct equivalent circuit for the particular corroding system presents one of the uncertainties of this method.

The interpretation of ECM data is based on a simplified corrosion model and the assumption of uniform corrosion, which is rarely encountered in magnesium and its alloys. Changes to the corroding surface induced by the polarization are the source of measurement artifacts. PDP evokes unusual phenomena such as negative difference effect (NDE). The obtained corrosion rate values are strongly dependent on the chosen measurement protocol and experimental setup, as well as on the mathematical interpretation of the results. 

In sum, none of the abovementioned methods alone provides a complete picture of the particular corrosion processes of the corroding system in question, nor are obtained values for corrosion speed necessarily comparable [9,33]. For this reason, a complementary use of these and additional methods is often recommended [2,31,33]. Further challenges arise through the choice of electrolyte and its chemical and physical parameters and the difficulties of replicating the conditions in the human body. Electrolyte components and parameters play a considerable role in determining corrosion speed, especially when electrolytes are chosen that approximate in vivo conditions [9,41]. Among other things, this is attributed to the fact that secondary reactions such as the production of insoluble compounds (phosphates, carbonates) and their interactions with chloride ions and enzymes are not properly taken into account. Additionally, physical conditions such as electrolyte temperature, pH value, flow rate, electrode surface-to-electrolyte ratio, immersion time, and overall experimental setup lead to large variations in reported corrosion rates [22,23,27,30,42]. Different electrochemical techniques were developed with the intention of overcoming these limitations. For example, the development of the Minicell [39] and Scanning Electrochemical Microscopy (SECM) in lieu of the classical cell permits a reduction of the area of analysis and, hence, a higher local resolution [36]. This miniaturization, in turn, raises the question to what extent the results can be extrapolated to the larger surfaces of real components such as stents and orthopedic screws. 

These inaccuracies in measurement and interpretation lead to the consequence that results from different in vitro corrosion experiments do not necessarily agree nor can they accurately predict the in vivo corrosion rates of magnesium alloys [22,31,43]. This has been demonstrated by comparing the predicted degradation rates derived from in vitro experiment to those actually achieved in animal studies [23]. In the most studies, in vitro measurement methods overestimate the actual in vivo degradation rate 

In sum, the experimental parameters that most adequately reproduce the in vitro conditions are not known. However, some agreement exists that the experimental setup needs to simulate the in vivo application as fully as possible [32]. Additionally, the informative value and validity of electrochemical measurements for the assessment of the degradation of magnesium (-alloys) and other biometals is strongly dependent on the possibility of correlation of measurements to the surface events actually occurring. So far, this is mainly attempted by examining the surface after corrosion measurement has taken place, ignoring the fact that time-dependent processes may be obscured. 

As the aim of the present study, both of these shortcomings were addressed through the development and the evaluation of a specialized miniature electrochemical cell that permits the real-time microscopic evaluation of a small (<1 mm^2^), well-defined surface area during electrochemical measurement, with video-microscopic observations of phenomena such as corrosion product deposition and hydrogen evolution. The insights derived from this method can be used to better understand and adapt conventional electrochemical measurement methods and corrosion models, by linking electrochemical measurement to physical surface phenomena, especially over time. This can improve their reproducibility and predictive value, thus leading to the reduction of the in vitro/in vivo gap.

## 2. Materials and Methods

A graphical abstract of the Materials and Methods section is provided in Figure 1.

Specimens of two different magnesium purity grades, commercial pure (cp) and extra-high pure (XHP), were used. 

The cp magnesium was cast and provided by Helmholtz Zentrum Geesthacht (Geesthacht, Germany) and the XHP magnesium was cast and provided by ETH Zürich (Switzerland). The resulting rod-shaped bulk material was cut into discs of 10 mm in diameter and 1.5 mm in thickness. One such disc of each material was embedded in cold-curing poly (methyl methacrylate) resin (Technovit 4004, Heraeus Kulzer GmbH, Wehrheim, Germany) using cylindrically shaped molds, leaving only one circular surface of the disc exposed. The reverse side of the embedding material was opened up by drilling and the specimen connected with a small quantity of amalgam to a short length of copper wire, allowing the conductive connection of the specimen to the working electrode terminal of a potentiostat. 

The specimens were ground on a rotary grinding and polishing machine (Struers Dap-V, Struers GmbH, Crinitz, Germany) with silica carbide sandpaper in steps of 500, 800, and 1200, and 2400 grit (WS Flex 18 C, Hermes Schleifmittel GmbH, Hamburg, Germany), using a rotation speed of 150 rpm; light, manual pressure; and water cooling. Each grinding step was continued for 5 min. Subsequently, the specimens were polished using a neoprene polishing plate (Mambo Chem, Industrieservice Siegmund Bigott, Kaarst, Germany) and a mixture of ethane diol-based colloidal silica dioxide suspension (OPS Extra, Industrieservice Siegmund Bigott, Kaarst, Germany) and alcohol-based 1 µ diamond spray (Diamantspray 1 µ, Industrieservice Siegmund Bigott, Kaarst, Germany) for 10 min, applying light, manual pressure. After polishing, the specimen surfaces were rinsed using pure ethanol and subsequently dried with pressurized air. 

To visualize grain structure, the polished specimens were etched with a solution of picric and acetic acid for about 1 s, before being rinsed with ethanol and dried with pressurized air [44]. The specimens were stored in a vacuum desiccator. 

Grain structure was recorded using a digital microscope (Keyence VHX-5000, Keyence Corporation, Osaka, Japan).

The specimens were polished once more according to the above protocol to prevent the remnants of the etching process from influencing the measurements.

The 3D areal surface texture of five randomly chosen rectangular areas of about 0.38 mm^2^ per specimen was captured using an optical measurement system (Alicona InfiniteFocus, Alicona Imaging GmbH, Raaba, Austria) at 20-fold magnification. For surface roughness characterization, the value for Sa (absolute arithmetical mean height according to ISO 25178) was calculated from the recorded 3D images with the aid of the equipment manufacturer’s software (Alicona IF-MeasureSuite 4.2).

For the electrochemical measurements and concurrent microscopic observation of the corroding surfaces, an opto-electrochemical cell was designed and constructed from Plexiglas (Evonik Röhm GmbH, Darmstadt, Germany). The miniaturized, electrochemical cell consisted of a chamber with a height of 7 mm and a contact area of about 0.8 mm^2^ to the bare surface of the specimen. Furthermore, the chamber had two electrolyte ducts, one of which connected to an electrolyte reservoir for filling the cell with electrolyte at the beginning of the experiment and the other connected to the Mini-Cell System (MCS) (Figure 2). The Mini-Cell System [45] contained the platinum counter electrode (CE) and the saturated calomel reference electrode (SCE, E_0_ = 0.241 V). 

Connected to the MCS, a pump allowed for the aspiration and optional cycling of the electrolyte through the optical-electrochemical cell and MCS chamber. The MCS was connected to a mini-potentiostat (PalmSens Emstat3+, PalmSens BV, Houten, The Netherlands) and the associated PC hardware running the potentiostat manufacturer’s software (PSTrace 5.3, PalmSens BV, Houten, The Netherlands). The top of the electrochemical cell consisted of a clear glass aperture of 2 mm in diameter, which allowed for direct observation of the corroding surface and associated phenomena with the aid of a microscope, in the present case, the digital microscope Keyence VHX-5000 (Keyence Corporation, Osaka, Japan).

This setup allowed for an observation of a well-defined circular area of about 1 square millimeter, undergoing corroding in contact with an electrolyte under a 200-fold microscopic magnification and simultaneous measurement and/or manipulation of voltage and current flow over the corroding surface. An example of the appearance of the magnesium surface as seen through the opto-electrochemical cell by microscope is shown in Figure 3.

As electrolyte, MEM (minimum essential medium) Earle’s w/o NaHCO_3_, L-Glutamine, and Phenol red (Biochrom GmbH, Berlin, Germany) was chosen to achieve an approximation of in vivo conditions [30] while establishing a baseline for the future addition of more complex components, such as fetal bovine serum (FBS) and additional buffer systems [43,46]. This electrolyte is among those commonly used for magnesium biodegradation measurements and results in relatively good correlations between in vitro and in vivo degradation rates [23]. It should be noted that electrochemical measurement results differed considerably according to the chosen electrolyte [47]. The components of the selected electrolyte according to the manufacturer’s data sheet [48] are listed in Table 1.

After filling the chamber with electrolyte, OCP was measured for 30 min. The OCP was assumed to be at steady state at the end of the 30-minute measurement period. This was followed by six voltammetric cycles with an amplitude of ±0.5 volt starting at a value of –1.6 V and with a scan rate of 10 mV/s. Hereby, the first vertex was cathodic and the second vertex was anodic. Simultaneously, a video of the corroding surface was recorded with a digital microscope (Keyence VHX-5000, Keyence corporation, Osaka, Japan). The resulting video was cut into sequences, and relevant stills for analysis were extracted. The obtained electrochemical data (OCP vs. time and I vs. E curves) were exported to a corrosion data analysis software (CView 3,0, Scribner Associates Inc., Southern Pines, NC, USA) and the values for OCP, electrochemical corrosion potential (E_corr_), linear polarization resistance (R_P_), and total anodic current (I_corr_) were extracted. Linear R_P_ was calculated from the reciprocal of the slope of a linear fit to the CV curve (defined as conductivity) over a segment of ±20 mV from E_corr_ multiplied by the exposed specimen surface and given in Ω∙cm^2^ (polarization resistance method). The corrosion current density (i_corr_) was calculated from R_p_ and the Stern–Geary constant according to [49,50,51].

For the purpose of this study, a Stern–Geary constant of 12.8 mV was assumed for the calculation of i_corr_ in agreement with published data on cp magnesium [51]. This value corresponds to an anodic and cathodic Tafel slope of 60 mV/decade. When comparing the obtained values for i_corr_ to those published in literature, it is important to keep in mind that in the present case they were calculated from R_P_ derived from linear polarization and an estimated Stern–Geary constant. However, reported values for B differ considerably in the case of magnesium.

Experiments were conducted in triplicate. Tabular data are presented as mean value ± standard deviation.

## 3. Results

The etched surfaces of the uncorroded surfaces of both magnesium qualities can be seen in Figure 4. The corresponding values of surface roughness, expressed as S_a,_ are listed in Table 2. 

Typical Open Circuit Potential (OCP) curves, obtained from cp Mg and XHP Mg measurement from corrosion onset up to 30 min after onset, are shown in Figure 5. As can be seen from these exemplary plots, for both magnesium grades, OCP generally started at potentials far catholically removed from the values at the 30-min mark. The cathodic OCP at the very beginning of the experiment, when the electrolyte first contacted the magnesium surface, corresponded with generalized production of hydrogen appearing as rapidly expanding and often detaching bubbles.

Within minutes, the hydrogen evolution was observed as coming to a near standstill, while at the same time the formation of a thin corrosion layer of dark appearance was noticed. The corrosion layer appeared less dense in the vicinity of adhered hydrogen bubbles. Upon the occasional detachment or confluence of adhered bubbles at this stage, an instantaneous cathodic drop in OCP value was recorded, momentarily disturbing the OCP equilibrium and initiating a renewed convergence toward the steady state OCP, as exemplified in Figure 5. This corresponded with the exposure of previously uncorroded areas where the particular bubble had been in direct contact with the specimen surface. Neither in the case of cp magnesium nor XHP magnesium was an onset of pitting corrosion observed within the duration of the measurement nor could a corresponding change in OCP value be measured. At the end of the 30-minute measuring period and after removal of adhered hydrogen bubbles the corrosion layer appeared relatively uniform with less dense areas in the vicinity of spots of hydrogen evolution, as can be seen in Figure 5. At the center of each film-free area of hydrogen evolution a dark spot of corrosion product formed. Evident from this observation was the fact that the locations of hydrogen evolution did not shift during the measurement period.

The obtained OCP values upon conclusion of the open circuit measurement after 30 min are given in Table 3 and Table 4 and compared graphically in Figure 6. OCP values for static conditions at ambient temperature did not differ notably from those under dynamic conditions.

A typical cyclic voltammetry (CV) plot is reproduced in Figure 7 using the example of cp Mg. In the interest of clarity, only one of the six cycles is reproduced. In general, the curve form and the corresponding key figures such as E_corr_ and R_p_ only varied slightly between cycles and adhered to the general characteristics presented hereafter. 

The cathodic part of a cycle was marked by generalized hydrogen evolution, which increased in intensity approaching the lower vertex, indicating the cathodic partial reaction of water electrolysis, and decreased as the potential approached the first E_corr,_ in close accordance with the measured electron flow. 

Aside from the evolving hydrogen (Figure 5), no changes in surface appearance were observed during the cathodic cycle. Coming from the cathodic vertex, the hydrogen evolution stopped approaching E_corr_ and remained that way for the beginning of the anodic part. This E_corr_ value, defined as the potential at the first zero current, is henceforth referred to as E_corr_(forward). At the same time, the measured electron flow increased only slightly with increasing anodic potential, both phenomena indicating the formation of a passive film. The anodic part of the cycle was characterized by the abrupt onset of pitting corrosion. The time of onset of pitting corrosion fluctuated considerably and no obvious relation to experiment parameters was evident. Pitting corrosion was extremely localized, often taking place at a single site. Video-optically, pitting corrosion was associated with intense hydrogen evolution from the pitting site. The pitting site itself presented as a slowly expanding dark front of deposited corrosion product.

In the case of cp magnesium, these spots expanded outward from the site of initiation along the surface preceded by a localized front of violent hydrogen evolution. In the case of XHP Mg, the hydrogen evolution was more strictly localized and associated with the development of pits surrounded by volcano-like depositions of corrosion product (Figure 8). 

Localized corrosion coincided with a sharp increase in electron flow. Once initiated, localized hydrogen evolution continued well into the following anodic cycle. Localized anodic corrosion resulted in a cathodic shift of E_corr_ on the back scan (E_corr_ (back)) as well as increase of corrosion current density (i_corr_ (back)). In the subsequent cathodic polarization, generalized hydrogen evolution could be once again observed; but, additionally, hydrogen evolution centered on the areas of anodic product deposition was preferential, without a change in appearance of these regions, indicating catalyzed cathodic hydrogen evolution [52,53,54]. The areas where localized corrosion took place re-passivated during the following cathodic polarization. The onset of localized corrosion during the subsequent anodic cycle was in every case at a different localization.

Apart from the described anodic localized corrosion (mode I), another anodic behavior could be observed, that of anodic passivation (mode II). In the case of the absence of anodic localized corrosion, no hydrogen evolution or product formation took place during the anodic part of the cycle. Correspondingly, no increase in current density took place; rather, the forward scan was retraced by the back scan. Accordingly, in these cases anodic polarization did not result in an increase in corrosion current density or a cathodic shift of E_corr_ (back).

Both XHP and cp magnesium fluctuated between these two behaviors from cycle to cycle, independent of obvious experimental parameters. At times, hybrid forms could be detected.

The measured values for E_corr_ (forward) and E_corr_ (back), as well as i_corr_ (forward) and i_corr_ (back), are listed in Table 3 and Table 4 and represented in Figure 9 and Figure 10. E_corr_ (back) and i_corr_ (back) refer only to those cases when localized anodic corrosion took place (mode I). The values obtained in dynamic conditions at physiological temperature displayed less variation than those obtained in static conditions at ambient temperatures. This might be due to the fact that electrolyte depletion effects and surface obstruction through hydrogen bubble attachment introduced a random factor under static conditions that was ameliorated under dynamic conditions. 

## 4. Discussion

The chosen measurement protocol utilizing the described optical-electrochemical cell revealed similar behavior across the tested XHP and cp magnesium specimens. These were localized corrosion during anodic parts of a cycle and generalized hydrogen evolution devoid of corrosion during cathodic parts of a cycle. Also, enhanced catalytic activity with anodic polarization could be observed; hereby, the corrosion products deposited during anodic part of a cycle became the preferred spots for hydrogen evolution [52,53,54]. Furthermore, it was shown that the complete or partial passivation of the magnesium surface through cathodic polarization led to a delay or prevention of anodic localized corrosion. The so-called negative difference effect (NDE) was also observed, this being an increase in hydrogen formation as a function of increasing anodic potential at the breakthrough potential. In cases where this effect did not appear, no cathodic shift of the corrosion potential was observed. This can possibly be used as a proof of the hydrogen accumulation realized by production of MgH_2_, which was responsible for the anodic shift of the corrosion potential E_corr_ [43].

During corrosion of magnesium, the following reactions are possible:(5)$$\mathrm{Mg}+2H_{2}O\to \mathrm{Mg}{\left(\mathrm{OH}\right)}_{2}+H_{2}$$
(6)$$2\mathrm{Mg}+2H_{2}O\to \mathrm{Mg}{\left(\mathrm{OH}\right)}_{2}+{\mathrm{MgH}}_{2}$$
(7)$$\mathrm{Overall}:\;3\mathrm{Mg}+4H_{2}O\to 2\mathrm{Mg}{\left(\mathrm{OH}\right)}_{2}+H_{2}+{\mathrm{MgH}}_{2}$$

With the help of these equations, the observations and results can be explained. During OCP measurement, here for a duration of 30 min, the initial hydrogen evolution was clearly visible. At the same time, the development of a surface layer, consisting of Mg(OH)_2_, was observed, which acted as a barrier for further fast degradation of Mg [55]. The OCP was based on the galvanic coupling of anodic and cathodic reactions over the exposed surface, forming a mixed potential, initially described by reactions (1) and (2) and later on by (5) and (6), which caused an anodic shift of the OCP. The value of OCP depended on the relationship between the current densities of the abovementioned reactions and/or the relationship between the active surface areas of the anodic and cathodic reactions. This can be used to explain the differences seen in OCP data between XHP Mg and cp Mg. The more cathodic shift in OCP of cp Mg was caused by catalytic support of reaction (6) [52].

The subsequent cyclic voltammetry disturbed these equilibriums and produced new data for the interpretation of the degradation XHP and cp Mg. The higher cathodic currents in the case of cp Mg revealed the higher catalytic activity of cp Mg in comparison to XHP Mg, as described by Birbilis et al. [53]. In the voltammetric forward scan, the MgH_2_ creation in direct contact with Mg(OH)_2_ was seen resulting in an anodic shift of E_corr_ of the forward scan. There were significant differences between XHP Mg and cp Mg, which were seen in the more anodic shift of E_corr_ forward, in the higher i_corr_ values, and its reproducibility. This demonstrated the higher activity, which was possibly caused by the impurities in cp Mg. An explanation for the scattering of the CVs of XHP Mg cannot be given at the moment, but the observations described in [55] are supported.

Utilizing potentiodynamic techniques to determine magnesium corrosion rate is not universally accepted in the science community [56]. This is due to incomplete knowledge and unsupported simplifications regarding the magnesium corrosion process such as the assumption of uniform corrosion. Because of parasitic chemical/electrochemical reactions and coverage effects of unknown magnitude occurring alongside the theoretical corrosion reaction, the calculated values for corrosion current density may not accurately reflect the actual corrosion rate [32,57], in particular, when using the Tafel equation for corrosion rate estimation [33]. Nevertheless, potentiodynamic methods can be employed to show the effects of experimental parameters and alloying on the cathodic and anodic branches [32].

The described measurement cell brings together a number of benefits. Among those is an easy, well-defined, and reproducible setup with minimal destruction to the sample surface, permitting repeated measurement on the same sample. Furthermore, the test conditions can be adjusted to approximate in vivo conditions.

Apart from the measurement protocols utilized in this study, further electrochemical protocols are possible without changing the setup. These include electrochemical impedance spectroscopy (EIS), linear sweep potentiometry, and long-term open circuit potentiometry. Depending on the research question, a number of parameters can be chosen or adjusted. These are perfusion rate, electrolyte composition and temperature, buffer system, electrolyte temperature, and specimen composition as well as measurement area. This way the often-unclear influence of the abovementioned factors on the measured results can be isolated. If a more pragmatic approach is needed, the parameters can be adjusted as far as possible to in vivo conditions.

The specimen composition in consideration was not limited to magnesium and its alloys. On the contrary, valuable insights can potentially be gleaned by utilizing the described measuring cell on other metallic biomaterials of interest, such as iron or zinc.

## 5. Conclusions

The described method consolidated versatility with reproducibility and brought together electrochemical data with the underlying optical surface phenomena in real time. Hydrogen evolution, corrosion product formation, and voltage/current can be correlated topographically and temporally.

These are the first investigations that show the complexity of such correlation, which are given here descriptively first. 

Further measurements with different parameters (different alloys, electrolyte composition, temperature, pH, surface treatment, and measurement protocol) are possible and necessary to get more quantitative results where the relation of surfaces changes over time are involved into the assessment of the electrochemical data.

## Figures and Tables

**Figure 1 materials-14-01601-f001:**
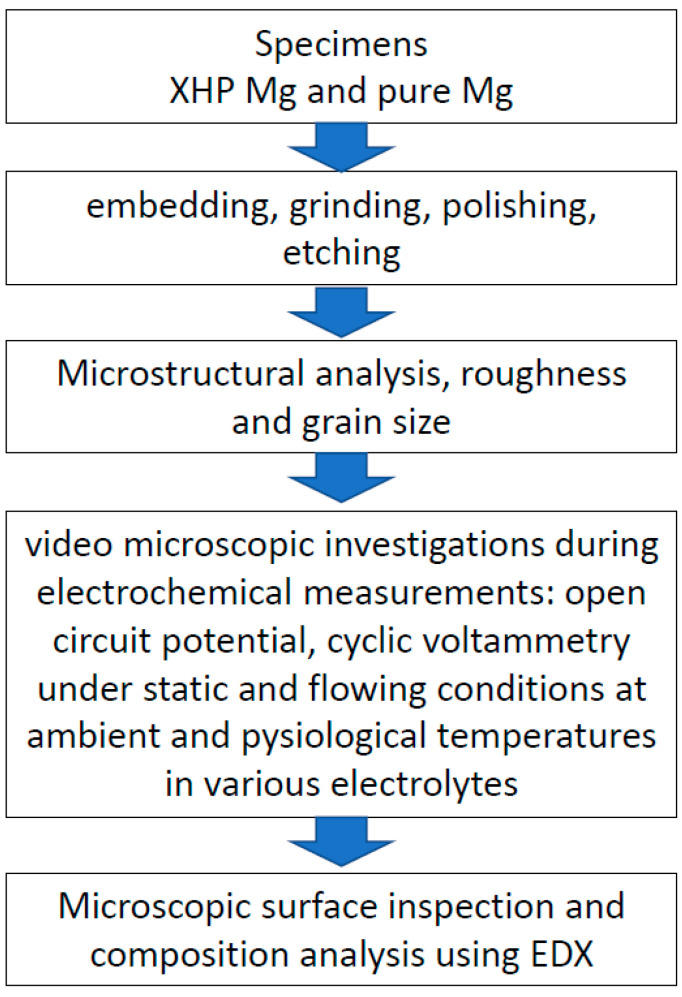
Graphical abstract of Material and Methods.

**Figure 2 materials-14-01601-f002:**
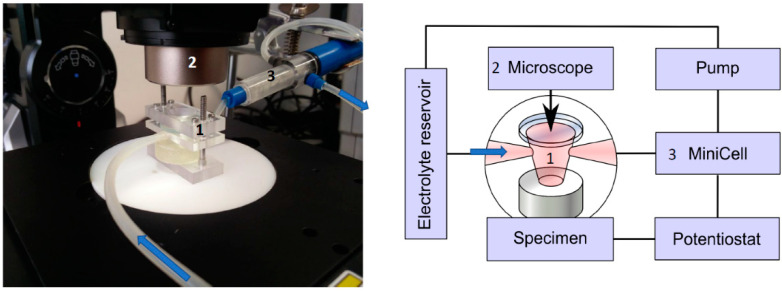
Video-optical-electrochemical cell with mounted specimen, connected to electrolyte inlet and Mini-Cell (background), beneath Keyence optical microscope. At left, a schematic view.

**Figure 3 materials-14-01601-f003:**
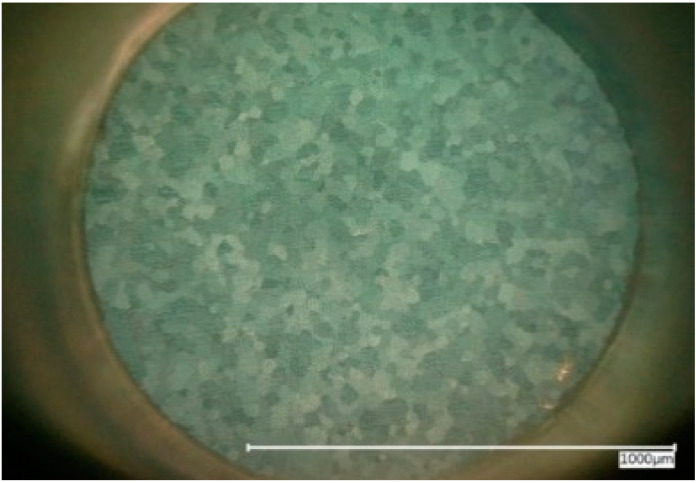
The cp Mg surface as seen through video-optical-electrochemical cell, polarized lighting.

**Figure 4 materials-14-01601-f004:**
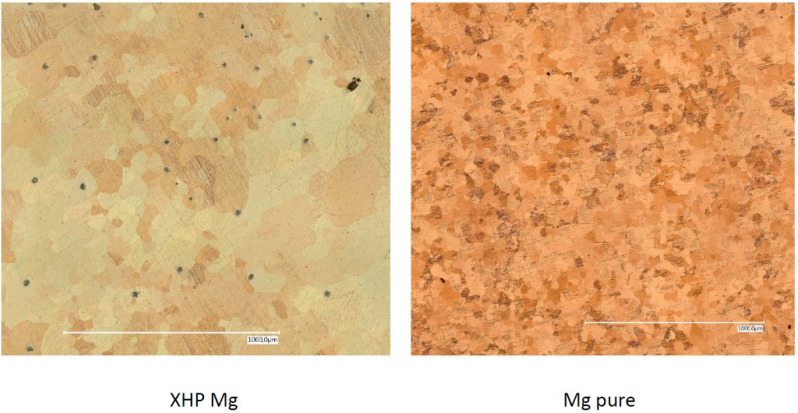
Microscopic appearance of etched surfaces of XHP (left) and cp (right) magnesium.

**Figure 5 materials-14-01601-f005:**
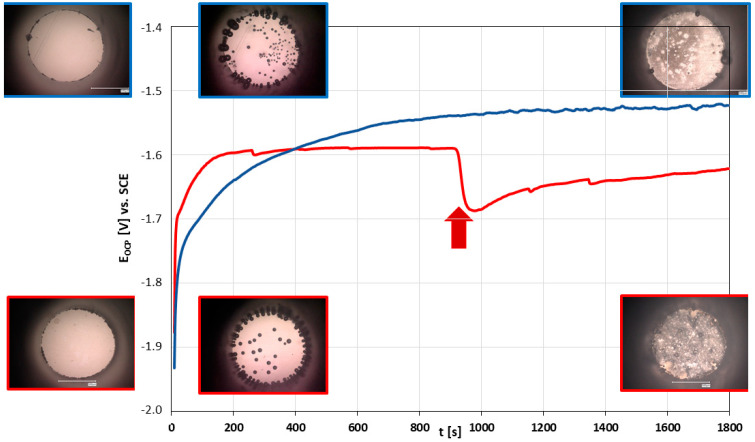
E_OCP_ vs. t plots for XHP (red) and cp (blue) magnesium aligned with microscopic pictures (prior to contact with electrolyte, as well as at t = 0 and 30 min, after removal of the hydrogen bubbles); arrow: spontaneous detachment of large hydrogen bubble exposing fresh magnesium surface.

**Figure 6 materials-14-01601-f006:**
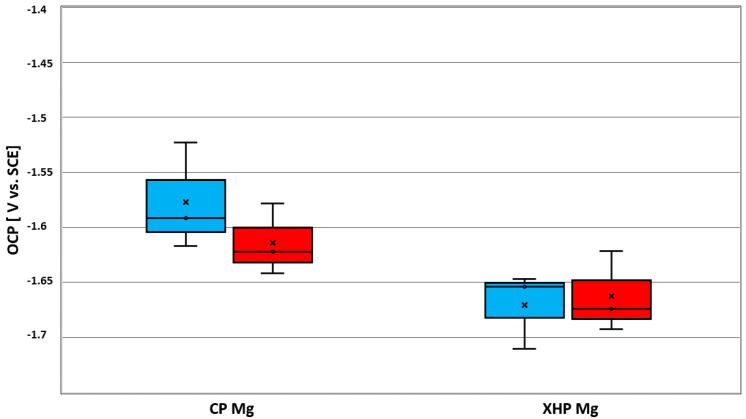
E_OCP_ after 30 min of XHP and cp magnesium in MEM at stationary and dynamic electrolyte flow conditions, at 25 °C. Left box relates to stationary and the right box to dynamic conditions.

**Figure 7 materials-14-01601-f007:**
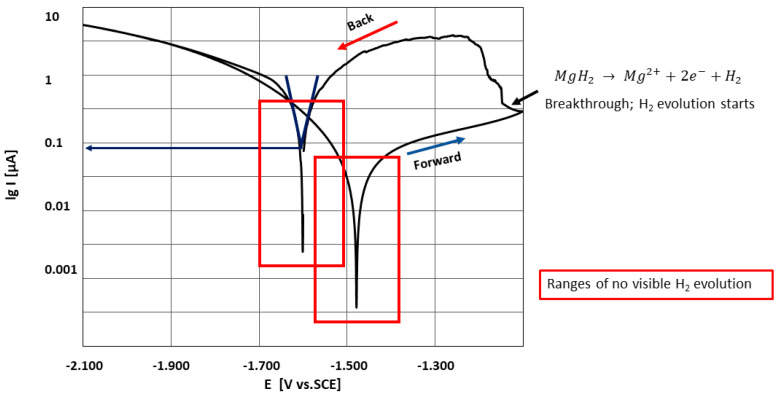
The principal shape of lgi vs. E curve. The red frames indicate the range where no hydrogen evolution was observed.

**Figure 8 materials-14-01601-f008:**
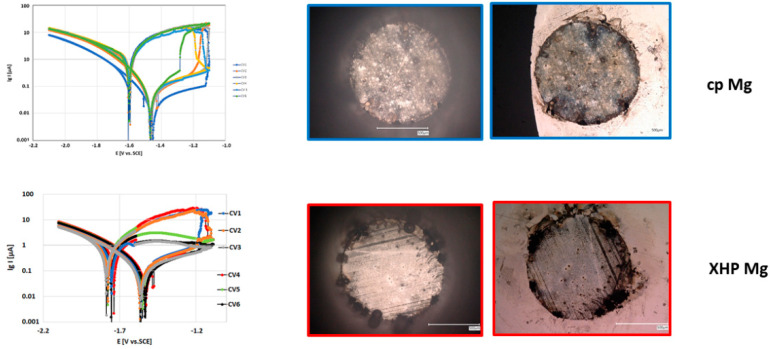
Left: lgi vs. E curves, cycle 1 and 6; right: microscopic view onto the surface after cyclic loading, (**left**) immediately after polarization and (**right**) after drying the surface. The upper row shows the CVs and microscopic pictures for cp Mg and the lower row for XHP Mg.

**Figure 9 materials-14-01601-f009:**
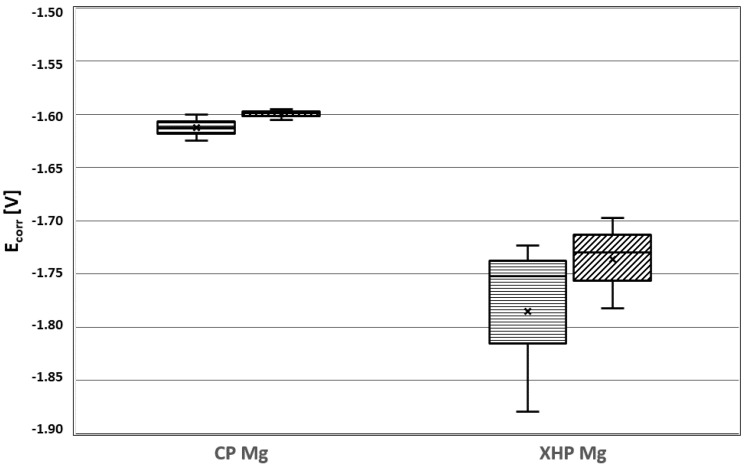
E_corr_ for static and dynamic conditions for XHP Mg and cp Mg in MEM.

**Figure 10 materials-14-01601-f010:**
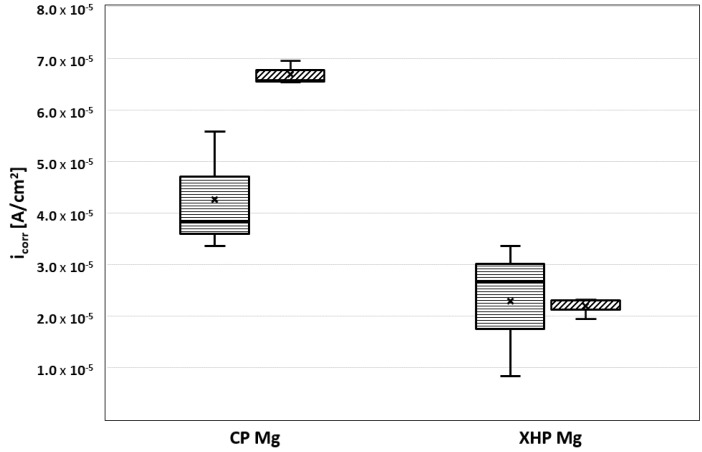
The i_corr_ for static and dynamic conditions for XHP Mg and cp Mg in MEM.

**Table 1 materials-14-01601-t001:** Electrolyte composition [49].

Substance	Concentration [mg/L]
Amino acids	1696
Vitamins & growth factors	8
NaCl	6800
KCl	400
NaH_2_PO_4_·H_2_O	140
MgSO_4_·7H_2_0	200
CaCl_2_	200
d-Glucose	1000

**Table 2 materials-14-01601-t002:** Surface roughness.

Name	Sa-Value [µm]
XHP Mg	0.10 ± 0.010
CP Mg	0.13 ± 0.012

**Table 3 materials-14-01601-t003:** Electrochemical measurement results, static conditions.

Name	E_ocp_ [V]	E_corr_ (Forward) [V]	E_corr_ (Back) [V]	i_corr_ (Forward) [A/cm^2^]	i_corr_ (Back) [A/cm^2^]
XHP Mg	−1.671 ± 0.035	−1.61 ± 0.13	−1.78 ± 0.07	3.5210^−6^ ± 1.8010^−6^	2.2810^−5^ ± 1.0610^−5^
CP Mg	−1.577 ± 0.049	−1.47 ± 0.02	−1.61 ± 0.01	2.9710^−6^ ± 1.78^−6^	4.2610^−5^ ± 9.5710^−6^

**Table 4 materials-14-01601-t004:** Electrochemical measurement results, dynamic conditions

Name	E_ocp_ [V]	E_corr_ (Forward) [V]	E_corr_ (Back) [V]	i_corr_ (Forward) [A/cm^2^]	i_corr_ (Back) [A/cm^2^]
XHP Mg	−1.663 ± 0.037	−1.50 ± 0.05	−1.74 ± 0.04	3.2610^−6^ ± 1.0810^−6^	2.1910^−5^ ± 1.7010^−6^
CP Mg	−1.614 ± 0.032	−1.48 ± 0.01	−1.60 ± 0.00	4.2410^−6^ ± 1.37^−6^	6.6910^−5^ ± 1.9110^−6^

## Data Availability

Not applicable.

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
