# Peer review of "The Video Microscopy-Linked Electrochemical Cell: An Innovative Method to Improve Electrochemical Investigations of Biodegradable Metals"

_materials, 2021, doi:10.3390/ma14071601_

Round 1

Reviewer 1 Report

The present manuscript can be accepted after the minor revision with addressing the following points.

  1. Briefly explain the effect of electrolyte on magnesium degradation
  2. References should be minimize up to 40 with the latest 
  3. Figure formate should be rearranged, especially for graphs (figure 6 and 9)
  4. Remove some typo errors.

Author Response

Dear reviewer thank you for Your time and input. Following your instructions, not only the mentioned but all figures have been made more legible. The text for the figures were supplemented and expanded. The explanation for the use of MEM as an electrolyte is described in Material and Method, Page 7 line 258 – 266.

The conclusions were supplemented with the note that these are the first descriptive results with the new measuring cell and that further investigations are required for quantification.

The typo errors were removed.

The reduction of the references to a maximum of 40 is difficult. Only the quotation [44] that was unrelated was removed and the following references renumbered accordingly.

We hope to fulfill the recommendations.

With kind regards

WD Mueller

(for the authors)

Reviewer 2 Report

The manuscript presents interesting data and could be published in the Journal,

nevertheless, several moments could be improved:

  1. The values in text, tables and figueres should in decimal form: 1*10^n, not 1*En
  2. check the data and error values through the text: 0.10 ± 0.010198039 is not correct as vell as for the ocp 1.504 in one case and 1.50 in another
  3. Redraw figs 7, 10, 11 - remove horizontal lines, header. Increase the font size, provide V vs (sce, nhe). Place the data rows closer to each other.

Author Response

Reviewer 2

Dear reviewer thank you for Your time and input. Following your instructions, not only the mentioned but all figures have been made more legible. The text for the figures were supplemented and expanded.

In the text, tables we change the exponential form into the decimal one. In case of the figures, where the current is presented the exponential values are still in the former form. We hope you can accept it.

The errors were checked and the mistake was changed, Table 2 between lines 302 to 303.

We hope to fulfill the recommendations.

With kind regards

WD Mueller

(for the authors)

Reviewer 3 Report

The author's purpose of the investigation is very interesting, also for scientists from related research fields. I would recommend the suggestions described below:
1)      The title should be short and concise. According to recent studies that would favor future citations.
2)      Abstract should be quantitative as possible for rapid comparison with others studies, referring quantitative values. After reading the paper some info is missing in the abs. The abs should be a mirror of the paper and not a kind of intro, aims or approaches. Insertion of a Fig 1 in the abs is absolutely  not normal.
3)      The paper includes 4 references from the last 4 years (but 0 from 2018, 2019 and 202). However, the authors should include references from the last 3 years in the field particularly if they claim for a new method and approach.

4)  Introduction is clearly random and it should be organized. Periods that are paragraph should be avoided. At the end of the intro, it also not clear what is the main message and relevant points of the paper that should be emphasize at this stage. What is really new and timely in the paper?
5)      The results are globally not properly described. The authors should first describe in a quantitative manner the data before jump to conclusions.  

6) Not all the figures are detailed described.
7)      The figures could be globally improve and normalized, once Materials deserves high quality figures and with rigor would avoid lacking of interest for the data. Type of letters in the figures should be normalized. For instance Figure 2 it not clear the correspondence of the photo with the drawing. Should it be added numbers for both?

8) Figures legends are clearly incomplete and need of clarifying all the abbreviations used. The letter in YY and XX are globally very small. No need to put the title above the graphics. This is not professional.

9) Discussion could be divided in section to clearly highlight the groups of results that needed to be emphasized.
10)      Globally the conclusions should followed the order of presentation of the paper with partial conclusions first and then global conclusions.

11) At the references section some parts are missing and even ref 44 only the year is mention.

Author Response

Reviewer 3

Dear reviewer thank you for Your time and input.

  • The title should be short and concise. According to recent studies that would favor future citations.

The Video-Microscopy Linked Electrochemical Cell – An Innovative Method to Improve Electrochemical Investigations of Biodegradable Metals

This is the new title of our paper.

  • Abstract should be quantitative as possible for rapid comparison with others studies, referring quantitative values. After reading the paper some info is missing in the abs. The abs should be a mirror of the paper and not a kind of intro, aims or approaches. Insertion of a Fig 1 in the abs is absolutely  not normal.

The abstract was changed into the following version and the figure was removed:

Abstract: An innovative miniature video-optical-electrochemical cell was developed and tested that allows for the conducting of electrochemical corrosion measurements and simultaneous microscopic observations over a small, well-defined surface area of corroding or degrading samples. The setup consists of a miniature electrochemical cell that is clamped onto the metal sample and fixed under a video microscope before being filled with electrolyte. The miniature cell comprises afferent/efferent electrolyte ducts as well as a connection to the Mini Cell System (MCS) for electrochemical measurements. Consequently, all measured and induced currents and voltages refer to the same small area corroding completely within the field of view of the microscope, thus allowing for real time observation and linking of surface phenomena such as hydrogen evolution and oxide deposition to electrochemical data. The experimental setup was tested on commercial purity (cp) and extra high purity (XHP) magnesium (Mg) samples using open circuit potential and cyclic voltammetry methods under static and flowing conditions.The corrosion potential is shifted more anodically for cp Mg in comparison to XHP Mg under dynamic conditions. The corrosion current assessed from the cyclic voltametric curves are higher for the cp Mg in comparison to XHP Mg. But there are no differences between static and flow conditions in case of XHP Mg in contrast to cp Mg, where the current density is two times higher at dynamic conditions.The measurements and observations with this new method pave the way for a more detailed understanding of magnesium corrosion mechanisms, thus improving predictive power of electrochemical corrosion measurements on newly developed magnesium or other biodegradable alloys applied for medical devices. Different electrochemical test can be run under various conditions, while being easy to set up and reproduce as well as being minimally destructive to the sample.

  • The paper includes 4 references from the last 4 years (but 0 from 2018, 2019 and 202). However, the authors should include references from the last 3 years in the field particularly if they claim for a new method and approach.

That is true.

  • Introduction is clearly random and it should be organized. Periods that are paragraph should be avoided. At the end of the intro, it also not clear what is the main message and relevant points of the paper that should be emphasize at this stage. What is really new and timely in the paper?

That we try to address better in a new version for the aim of the paper.

As the aim of the present study, both of these shortcomings are addressed through the development and the evaluation of a specialized miniature electrochemical cell that permits the real-time microscopic evaluation of a small (<1 mm2), well-defined surface area during electrochemical measurement,with video-microscopic observations of phenomena such as corrosion product deposition and hydrogen evolution. The insights derived from this method can be used to better understand and adapt conventional electrochemical measurement methods and corrosion models, by linking electrochemical measurement to physical surface phenomena, especially over time. This can improve their reproducibility and predictive value, thus leading to the reduction of the in vitro/in vivo gap. Lines 171 to 184.

5.) The results are globally not properly described. The authors should first describe in a quantitative manner the data before jump to conclusions.  

These is not easy because of the complexity of the data set we received. Therefore, we decide to publish our first results using quantitative data with a parallel description of the visualization in the videos therefore we re looking for a new way for publication. Concerning the connection of time, surface picture, electrochemical data to came to a clear conclusion needs a bit more time and capacity (man power). We hope you understand it, when we try to deliver the data for other groups to play and deal with.

6) Not all the figures are detailed described.
7)      The figures could be globally improve and normalized, once Materials deserves high quality figures and with rigor would avoid lacking of interest for the data. Type of letters in the figures should be normalized. For instance Figure 2 it not clear the correspondence of the photo with the drawing. Should it be added numbers for both?

8) Figures legends are clearly incomplete and need of clarifying all the abbreviations used. The letter in YY and XX are globally very small. No need to put the title above the graphics. This is not professional.

Following your instructions all figures have been made more legible. The text for the figures were supplemented and expanded.

9) Discussion could be divided in section to clearly highlight the groups of results that needed to be emphasized.
10)      Globally the conclusions should followed the order of presentation of the paper with partial conclusions first and then global conclusions.

We did not change the discussion because of the descriptive style, as we tried to explain you above.

The conclusions were supplemented with the note that these are the first descriptive results with the new measuring cell and that further investigations are required for quantification.

We hope to fulfill mostly your recommendations.

With kind regards

WD Mueller

(for the authors)

Round 2

Reviewer 3 Report

The authors were very positive, the answers fully comprehensive and the paper was globally improved.